# Integrated Pan-Cancer Analysis and Experimental Verification of the Roles of Retinoid-Binding Proteins in Breast Cancer

**DOI:** 10.3390/cancers17223706

**Published:** 2025-11-19

**Authors:** Yuchu Xiang, Dan Du, Yaoxi Su, Linghong Guo, Siliang Chen

**Affiliations:** 1Department of Dermatology & Venereology, West China Hospital, Sichuan University, Chengdu 610041, China; xiangyuchu@stu.scu.edu.cn (Y.X.); dudan.meishan@foxmail.com (D.D.); suyaoxi@wchscu.edu.cn (Y.S.); 2Laboratory of Dermatology, Clinical Institute of Inflammation and Immunology, Frontiers Science Center for Disease-Related Molecular Network, West China Hospital, Sichuan University, Chengdu 610041, China

**Keywords:** retinoid-binding proteins, pan-cancer analysis, RBP4, RBP7, breast cancer, tumor microenvironment

## Abstract

Many cancers disrupt how vitamin A-related proteins control cell growth and death. We studied these proteins, especially RBP4 and RBP7, to learn how their levels differ across cancers and what that means for patients, with special attention to triple-negative breast cancer, a difficult-to-treat subtype. Using large cancer datasets and single-cell analysis, we mapped where these proteins are reduced, how they change with tumor stage and grade, and how they relate to survival. We found that RBP7 shows patterns linked to better outcomes in several cancers, including breast cancer. Our goal is to identify which patients could benefit from using these proteins as markers to predict prognosis and to guide the design of new treatments. These results give researchers a clearer picture of the tumor environment and provide testable targets for future laboratory and clinical studies.

## 1. Introduction

Breast cancer is the most commonly diagnosed cancer and the leading cause of cancer-related deaths among women globally [1]. Current treatment options include surgery, radiotherapy, chemotherapy, endocrine therapy, and targeted therapy [2]. Although these strategies have improved survival and prognosis, important problems persist. Treatment resistance, adverse effects, and marked interpatient variability still limit long-term outcomes [3,4]. Thus, there is an urgent need to identify novel prognostic biomarkers and molecular targets to better predict outcomes and guide individualized therapy for patients with breast cancer. Retinoid-binding proteins (RBPs) are a crucial family of proteins involved in the transport and metabolism of retinoids, which are derivatives of vitamin A. These proteins play essential roles in cellular processes such as differentiation, proliferation, and apoptosis [5]. Among them, RBP4 is particularly notable for its role in transporting retinol (vitamin A alcohol) from the liver to peripheral tissues, ensuring adequate retinoid availability for various physiological functions [6]. RBP3, also known as interphotoreceptor RBP, is distinct from the intracellular RBPs, and is involved in retinoid transport between cells in the retina, contributing to the visual cycle [7]. RBP7, another important member of the RBP family, is gaining attention for its potential role in tumor biology. Emerging evidence suggests that RBP7 may be involved in cancer development and progression, though its exact mechanisms remain under investigation [8,9]. Other proteins in the RBP family, such as RBP1, RBP2, RLBP1, and the cellular retinoic acid-binding proteins (CRABP1 and CRABP2), contribute to retinoid metabolism and signaling but are less emphasized in this context.

In the context of cancer, RBPs have been shown to influence tumor growth and progression through their regulation of retinoid signaling pathways. Dysregulation of these proteins can lead to aberrant retinoid metabolism and signaling, contributing to oncogenesis and cancer progression [8,9]. For example, altered expression of RBP4 has been associated with poor prognosis in several cancers, including breast cancer [10], gastric cancer [11], and hepatocellular carcinoma [12]. Similarly, CRABP2 has been linked to the modulation of retinoic acid signaling, affecting cell differentiation and apoptosis in cancer cells [13,14]. Dysregulation of RBP7 expression in certain cancers hints at its potential influence on tumor biology, yet comprehensive studies are required to uncover its specific functions and therapeutic relevance in cancer contexts. Recent research has highlighted RBP7’s potential role in breast cancer, drawing attention to its influence on tumor biology and patient prognosis [8,15,16].

Bioinformatics analysis has emerged as a powerful tool for decoding the complexities of cancer genomics and transcriptomics. By integrating and analyzing large-scale datasets, researchers can identify key molecular players and pathways involved in cancer development and progression. In this study, our primary focus was on exploring the structural characteristics of RBP members and conducting a comprehensive analysis of their expression levels and prognostic implications across various cancers using data from the Cancer Genome Atlas (TCGA) database. We also investigated the correlation between RBP expression levels and immune checkpoints. Additionally, single-cell RNA sequencing (scRNA-seq) was used to analyze the expression of RBP4 and RBP7 in triple-negative breast cancer (TNBC), revealing their distribution across various cell types within the tumor microenvironment. Our findings shed light on the detailed expression profiles and overall landscape of immune infiltration mediated by the RBP gene family in pan-cancer scenarios, thereby offering valuable insights into the potential of RBPs as therapeutic targets across diverse cancer types.

## 2. Materials and Methods

### 2.1. Analysis of the Key Features of RBP Family Members in Homo sapiens

In this study, phylogenetic relationships among RBP family members were inferred in MEGA v7.0.26 using the neighbor-joining (NJ) algorithm with 1000 bootstrap replicates [17], and the resulting tree was subsequently rendered in TBtools (v2.1) [18]. For structural analyses, GFF3 (General Feature Format Version 3) annotation files for RBPs were retrieved from the Ensembl database and used to examine gene organization. Conserved motifs within RBPs were characterized with the MEME suite (http://meme-suite.org/tools/meme, accessed on 23 July 2024). In addition, TBtools was employed to analyze the gene structures of all RBPs, and the SWISS-MODEL Interactive Workspace was applied to predict the three-dimensional (tertiary) structures of RBPs [19,20].

### 2.2. TCGA Pan-Cancer Atlas Data Profile

We accessed TCGA pan-cancer cohorts through UCSC Xena (https://xenabrowser.net/, accessed on 28 July 2024), obtaining data for RBP family members that included gene expression profiles, clinicopathological parameters, molecular subtype annotations, survival information, and related variables [21]. Expression matrices for the RBP family were merged using Perl scripts, and differences between tumor and matched normal tissues were evaluated with the Wilcoxon rank-sum test. Statistical significance was denoted as “*”, “**”, and “***” for *p* < 0.05, *p* < 0.01, and *p* < 0.001, respectively. The R packages “ggpubr” (v 0.6.0) and “pheatmap” (v 1.0.13) were applied to display RBP expression patterns using boxplots and heatmaps. Correlations among RBP family genes were examined with the “corrplot” package. Furthermore, “ggplot2” was used to explore associations between RBP expression and clinical stage and grade, and the GSCA platform was employed to assess the relationship between RBP family expression and molecular subtypes across different tumor types.

### 2.3. Survival and Cox Analysis of RBP Family Expression

Survival differences according to the expression of RBP family genes in each tumor type were examined using Kaplan–Meier curves and log-rank tests. The prognostic relevance of these genes was analyzed in R with the “survival” and “survminer” packages. In addition, univariate Cox proportional hazards regression was applied to quantify the association between RBP expression and prognosis across cancers, and the resulting hazard ratios were summarized in a forest plot generated with the “survival” (v 3.3.1) and “forestplot” (v 3.1.7) packages.

### 2.4. Genetic Alteration Analysis

The cBioPortal platform integrates data from comprehensive tumor genome studies, including large projects like TCGA and ICGC, encompassing data from over 28,000 samples [22]. In this study, we utilized the cBioPortal database to investigate the mutation frequency and forms of RBP family members.

### 2.5. Immune Infiltration Cells and Immune Checkpoint Correlation Analysis

In this study, we downloaded the standardized pan-cancer dataset from the UCSC (https://xenabrowser.net/, accessed on 28 July 2024) database: TCGA Pan-Cancer (PANCAN, N = 10,535, G = 60,499). From this dataset, we extracted the expression data of the RBP4/7 gene and 60 marker genes of two categories of immune checkpoint pathways (Inhibitory (24) and Stimulatory (36)) across various samples. We further selected samples originating from Primary Blood-Derived Cancer, Peripheral Blood, and Primary Tumor, excluding all normal samples. Each expression value was log2 (x + 0.001)-transformed. The associations between RBP family genes and selected representative immune checkpoints were analyzed using the Spearman correlation test. Significance levels were indicated as follows: * *p* < 0.05, ** *p* < 0.01, and *** *p* < 0.001.

### 2.6. Single-Cell Transcriptome Analysis

#### 2.6.1. Data Preprocessing

Matrix, barcodes, and features files of scRNA data were downloaded from GSE161529. Of these scRNA data, 5 normal breast samples and 4 TNBC samples were included. Following initial Cell Ranger processing, Seurat (v 4.0.0) was used for further quality control. Cells with >200 genes, >1000 UMIs, a log10GenesPerUMI > 0.7, <5% mitochondrial UMIs, and <5% erythrocyte gene expression were retained. Doublets were removed using DoubletFinder (v2.0.3), and data were normalized using Seurat’s NormalizeData function. Batch corrections were conducted using the “Harmony” method.

#### 2.6.2. Dimensionality Reduction and Clustering

Highly variable genes (top 2000) were selected using Seurat’s FindVariableGenes function. Principal component analysis (PCA) and uniform manifold approximation and projection (UMAP) were used for dimensionality reduction and clustering.

#### 2.6.3. Marker Gene Identification

The FindAllMarkers function in Seurat identified upregulated genes per cell type. Marker genes were visualized with VlnPlot and FeaturePlot.

#### 2.6.4. Cell Type Identification

Cells were manually annotated and checked using the top 10 marker genes identified (Appendix A). The percentage of RBP4-positive and RBP7-positive cells in each cluster was calculated and compared.

#### 2.6.5. Genes Co-Expressed with RBP7 and Functional Analysis

Using TCGA datasets accessed through the Xiantao platform, we ranked genes by their correlation with RBP7 and selected the 50 most positively and 50 most negatively co-expressed genes for heatmap visualization. A protein–protein interaction (PPI) network was then constructed for the 100 positively co-expressed genes using STRING (https://cn.string-db.org/, accessed on 14 August 2024), which compiles publicly available PPI information [23]. Putative hub genes within this network were identified with the “MCODE” and “cytoHubba” plug-ins in Cytoscape (version 3.7.2).

### 2.7. Tissue Microarray Methodology and Validation

A tissue microarray (TMA) was constructed using formalin-fixed, paraffin-embedded pathological specimens from 47 patients enrolled between 2005 and 2012. Representative tumor regions were marked, and 2.0 mm cores were extracted and arranged in a new paraffin block. Sections of 4 µm were cut, deparaffinized, and subjected to immunohistochemical staining for RBP7, using a specific primary antibody (Proteintech, 14541-1-AP) and a chromogenic detection system. The expression of RBP7 was assessed by two independent pathologists based on staining intensity and the percentage of positive cells. The follow-up period for these patients ranged from 31 to 132 months. In addition, the RBP7 expression and prognostic impact were validated using GENT2 (http://gent2.appex.kr/gent2/, accessed on 6 November 2025) [24], an online tool for investigating the gene expression profiles across tumor and normal tissues based on data from ArrayExpress database and GEO database, and the “Kaplan–Meier Plotter-Breast cancer” (https://kmplot.com/analysis/index.php?p=service&cancer=breast, accessed on 6 November 2025), an online tool for investigating the prognostic impact of RBP7 also based on the ArrayExpress and GEO databases [25].

### 2.8. Statistical Analysis

All statistical procedures in this study were performed using the aforementioned online platforms together with R (RStudio v1.2.1335, R v3.6.3). Experimental datasets were analyzed in GraphPad Prism 9.0 (GraphPad Software, La Jolla, CA, USA). Comparisons between groups were made with Student’s *t*-tests, and data are expressed as mean ± SD. Statistical significance was defined as follows: * *p* < 0.05, ** *p* < 0.01, *** *p* < 0.001, and **** *p* < 0.0001. For multiple testing, such as the DEG screening in pan-cancer analysis and marker gene identification in scRNA-seq analysis, *p*-values were corrected using the false-discovery rate (FDR) adjustment.

## 3. Results

### 3.1. Gene Structure and Motif Composition of RBP Family Members in Homo sapiens

Retinoid-binding proteins (RBPs) are a family of proteins that mediate the transport and metabolism of retinoids; they are derivatives of vitamin A that are essential for cellular differentiation, proliferation, and apoptosis [5]. In this study, we first characterized nine RBP family members in *Homo sapiens* using the UniProt and GeneCards databases. Most of these proteins were predicted to localize predominantly in the cytoplasm. Gene Ontology (GO) Terms obtained from the “Pathway” module of GeneCards further indicated that RBP family members are involved in diverse biological processes. For example, RBP4 and RBP7 are annotated with retinoid, retinal, and retinol binding, as well as protein-binding activity (Table 1). To further elucidate the phylogenetic relationships within the human RBP family, we constructed a phylogenetic tree using the sequences of RBPs in this study. The phylogenetic analysis of the RBP family revealed two major clades: one containing RBP3 and the other comprising all remaining RBPs. In the second clade, RBP4 and RLBP1 formed one subgroup, CRABP1 and CRABP2 formed another, while RBP1 clustered with RBP5 and RBP2 with RBP7. These relationships were supported by high bootstrap values, indicating strong evolutionary connections (Appendix A).

### 3.2. Expression Levels of the RBP Family in Pan-Cancer

Based on previous studies classifying the RBP family, we estimated the differential expressions of RBPs in pan-cancer using RNA sequencing data from the TCGA database, including RBP1, RBP2, RBP3, RBP4, RBP5, RBP7, RLBP1, CRABP1, and CRABP2.

Our findings revealed that RBP1 exhibited significant overexpression in esophageal carcinoma (ESCA), glioblastoma (GBM), head and neck squamous cell carcinoma (HNSC), and lung squamous cell carcinoma (LUSC), while showing marked decreases in bladder urothelial carcinoma (BLCA), kidney renal clear cell carcinoma (KIRC), kidney chromophobe (KICH), liver hepatocellular carcinoma (LIHC), kidney renal papillary cell carcinoma (KIRP), lung adenocarcinoma (LUAD), stomach adenocarcinoma (STAD), and prostate adenocarcinoma (PRAD) (Figure 1A). RBP2 exhibited significant downregulation across several tumors, including KICH, KIRC, KIRP, LUAD, LUSC, and thyroid carcinoma (THCA), with elevated expression observed in breast invasive carcinoma (BRCA), cholangiocarcinoma (CHOL), colon adenocarcinoma (COAD), and LIHC (Figure 1B). Similarly, RBP3 showed decreased expression in most tumor tissues, including BRCA, GBM, KICH, KIRC, KIRP, PRAD, THCA, and uterine corpus endometrial carcinoma (UCEC), while exhibiting increased expression in cervical squamous cell carcinoma and endocervical adenocarcinoma (CESC), bladder urothelial carcinoma (BLCA), COAD, pheochromocytoma and paraganglioma (PCPG), and lung squamous cell carcinoma (LUSC) (Figure 1C). Across tumor types, RBP4 expression was markedly lower in BLCA, BRCA, CHOL, GBM, HNSC, KICH, KIRC, LIHC, LUAD, LUSC, PCPG, and PRAD, whereas elevated expression was detected only in COAD and STAD (Figure 1D). Likewise, RBP5 exhibited significant downregulation in BLCA, CESC, CHOL, COAD, KICH, KIRP, LIHC, LUSC, PRAD, rectum adenocarcinoma (READ), THCA, and UCEC, but was upregulated in esophageal carcinoma (ESCA) and HNSC (Figure 1E). RBP7 exhibited decreased expression levels in BLCA, BRCA, CESC, COAD, HNSC, KICH, KIRP, LUAD, LUSC, PRAD, READ, STAD, THCA, and UCEC, with higher expression observed in KIRC and LIHC (Figure 1F). RLBP1 protein analysis revealed decreased expression in BRCA, COAD, LUAD, LUSC, and READ, and increased expression in HNSC, KICH, KIRC, KIRP, and UCEC (Figure 1G). CRABP1 expression analysis indicated significant decreases in BLCA, BRCA, COAD, HNSC, KICH, KIRC, KIRP, LIHC, PRAD, READ, and THCA, with increased expression observed in LUAD, LUSC, and UCEC (Figure 1H). Finally, CRABP2 expression was significantly increased in BRCA, CHOL, COAD, ESCA, LIHC, LUAD, LUSC, STAD, and THCA, while it was decreased in KICH, KIRC, KIRP, and PRAD (Figure 1I).

### 3.3. The Association Between Clinical Characteristics, Tumor Subtypes, and RBP Expression

Additionally, we analyzed the expression levels of RBP genes across different subtypes of nine cancers. As shown in Figure 2A, most RBP genes exhibited significantly different protein expression levels in Breast Invasive Carcinoma (BRCA). Furthermore, we found that the expression of RBP7 in four subtypes of BRCA (Basal, Her2, LumA, and LumB) was significantly different compared to the normal-like subtype. The expression of RBP4 and RLBP1 differed significantly in Her2, Basal, and LumB subtypes compared to normal tissue. The expression of RBP1, RBP3 and CRABP1 also differed significantly in Her2, LumA, and LumB subtypes compared to normal tissue. However, there were no significant differences in RBP2 expression levels between Basal, Her2, LumA, LumB, and normal-like subtypes (Figure 2B).

### 3.4. Prognostic Value of RBPs in Pan-Cancer

To explore the association between RBP family gene expression levels and prognosis in BRCA, we conducted survival analysis using the Kaplan–Meier survival curves. The results indicated that RBP1, RBP4, and RBP7 exert protective effects in BRCA (Figure 3A).

Subsequently, we explored the prognostic risk of RBP genes through univariate Cox analysis. As shown in Figure 3B, RBP4 acts as a protective gene in KIRP, LIHC, and mesothelioma (MESO), while it is a high-risk factor in HNSC, ovarian serous cystadenocarcinoma (OV), sarcoma (SARC), and UCEC. RBP7 plays a protective role in BRCA, KIRC, and uveal melanoma, but is a high-risk prognostic factor in COAD, and STAD.

### 3.5. Genetic Alteration and Immune Checkpoint Correlation Analysis

We further analyzed the variation frequency and types of the RBP gene family in 2922 cancer patients via the cBioPortal database. The results indicated that the mutation rate of the CRABP2 gene was approximately 13%, the highest among the RBP family. The mutation rates of RBP4, CRABP1, and RLBP1 were around 5%. However, RBP3 and RBP7 had the lowest mutation rates, only 4% each (Figure 4A).

Furthermore, we observed that the main types of genetic alterations included amplification, high mRNA expression, mutation, and deep deletion. CRABP2 exhibited the highest amplification rate. Meanwhile, RBP4 primarily showed high mRNA expression alterations, and RBP7 alterations included amplification and high mRNA expression. Mutations were predominantly observed in RBP3, RBP4, and RLBP1 (Figure 4B). In conclusion, members of the RBP gene family are prone to genetic alterations across various cancer types.

In addition, we conducted co-expression analysis using the TCGA database to reveal the association of RBP4 and RBP7 with immune checkpoints in pan-cancer (Appendix A).

### 3.6. Single-Cell Transcriptome Analysis of RBP4 and RBP7 in TNBC

Further analysis of single-cell transcriptome data revealed the expression profiles of RBP4 and RBP7 in cancers. As shown in Appendix A, each cell’s nFeature_RNA (number of genes), nCount_RNA (number of UMIs), and the proportion of mitochondrial genes relative to nFeature_RNA were assessed before filtering. After filtering based on gene expression counts, and mitochondrial gene proportions, the results in Appendix A were obtained. We observed a strong positive correlation between filtered nCount_RNA and nFeature_RNA, indicating good data quality suitable for further analysis (Appendix A). Following PCA dimensionality reduction, representative genes for PC1 and PC2 are shown in Appendix A, demonstrating the effective batch correction achieved using the Harmony method. Appendix A depicts UMAP clustering of cells into 11 clusters, illustrating their spatial distribution. Further analysis of the top 10 marker genes for these 11 clusters is presented in Appendix A, represented by a heatmap and a bubble plot, respectively.

After manual annotation and checking, we identified 11 cell clusters: basal cells, fibroblasts, tumor epithelial cells, T cells, luminal epithelial cells, endothelial cells, proliferating tumor cells, pericytes, macrophages, luminal secretory cells, and B cells (Figure 5A). Figure 5B shows the top 10 marker genes corresponding to each cell cluster after annotation. Further analysis of RBP4 and RBP7 expression in TNBC (triple-negative breast cancer) tissue revealed that RBP4 was mainly expressed in endothelial cells. While RBP7 was primarily expressed in endothelial cells, it was also expressed at low levels in tumor epithelial cells, proliferating tumor cells, and luminal epithelial cells. (Figure 5C–F). Furthermore, we calculated the percentage of RBP4- and RBP7-positive cells for each cell cluster (Appendix A) and compared them between TNBC and normal tissues. We found that the RBP4-positive cells are significantly decreased for basal cells, endothelial cells, and pericytes in TNBC (Appendix A), while RBP7-positive cells are significantly decreased for luminal secretory cells and pericytes and significantly increased for macrophages in TNBC (Appendix A).

### 3.7. Single-Cell Differential Gene Expression Analysis

In tumor cells, we analyzed the expression levels of the top 50 genes that were negatively and positively correlated with RBP7 expression. The heatmap results indicated that the majority of these genes were positively correlated with RBP7 expression, while only a few were negatively correlated (Figure 6A,B). Subsequent GO functional enrichment analysis revealed that the differentially expressed genes were primarily located in the extracellular region and vesicle (Figure 6C). KEGG pathway enrichment analysis indicated that these differentially expressed genes were mainly involved in PI3K-Akt pathways, ECM-receptor interaction, the IL-17 signaling pathway, leukocyte transendothelial migration, and glutathione metabolism (Figure 6D). Furthermore, we performed a protein–protein interaction (PPI) analysis of the top 100 significantly positively correlated genes (Figure 6E). Based on the PPI network, we used cytoHubba to select the top 10 and top 5 hub genes, which mainly included TYMS, TK1, ASF1B, UBE2C, and PCLAF (Figure 6F,G).

Next, we validated the expression and prognostic significance of RBP7 using tissue microarray and immunohistochemistry in pathological specimens from 47 breast cancer patients. Patient survival was found to be proportional to RBP7 expression levels, with higher RBP7 expression correlating with better prognosis (Figure 7A). The staining patterns of RBP7 in adjacent non-cancerous tissue and breast cancer tissue also indicated a significant reduction in RBP7 expression levels within the tumor tissue (Figure 7B,C and Appendix A). In addition, we also validated the expression profiles of RBP7 across tumor and normal tissues of breast cancer using the online tool GENT2, and validated the prognostic impact of RBP7 on breast cancer using the online tool “Kaplan–Meier Plotter-Breast cancer”. Both of these online tools were based on the expression data from the ArrayExpress and GEO databases. We found that RBP7 was significantly decreased (*p* < 0.001) in breast cancer (Figure 7D) and served as a protective factor (HR = 0.86, *p* < 0.05) for breast cancer survival (Figure 7E), which was consistent with the findings from the TCGA cohort and the immunohistochemistry staining of the tissue microarray performed in our study.

## 4. Discussion

RBPs play crucial roles in cancer biology by modulating retinoid metabolism and signaling pathways, influencing key cellular processes such as differentiation, proliferation, and apoptosis [8,9,10,11,12,13,14,15,16]. Our comprehensive analysis, using bioinformatics tools and experimental validations, revealed distinct expression profiles of RBP family members across various cancers, highlighting the critical role of RBP4 and RBP7 in the context of triple-negative breast cancer biology. In the pan-cancer analysis, we found that RBP1, RBP2, RBP3, RBP4, RBP5, RBP7, RLBP1, and CRABP1 were downregulated in the tumor group compared to the normal group across most of the cancer types in TCGA. However, CRABP2 expression in pan-cancer showed an opposite pattern. It is noteworthy that RBP4 showed significant decreases in BLCA, BRCA, CHOL, GBM, HNSC, KICH, KIRC, LIHC, LUAD, LUSC, PCPG, and PRAD, while RBP7 exhibited reduced expression across 14 types of tumors, namely, BLCA, BRCA, CESC, COAD, HNSC, KICH, KIRP, LUAD, LUSC, PRAD, READ, STAD, THCA, and UCEC, which may have been one of the reasons affecting the outcomes of these cancers. Furthermore, we examined RBP expression patterns across tumor stages and histological grades. Most RBP family genes showed marked changes in expression during cancer progression, leading us to propose that these genes may serve as useful diagnostic and prognostic indicators in pan-cancer settings. Although previous researchers had investigated RBP7 in breast cancers, our study newly explored the role of RBPs across all cancer types and highlighted the role of RBP4 and RBP7 in TNBC.

Through Cox analysis, we found that RBP4 acts as a protective gene in KIRP, LIHC, and MESO, while RBP7 displayed protective effects in BRCA and UVM. By summarizing the expression levels of RBP genes across nine cancer types, we observed that, particularly in BRCA, most RBP genes showed notable variations in protein expression. Furthermore, survival analysis revealed that RBP1, RBP4, and RBP7 exert protective effects in BRCA. However, the role of RBP4 in breast cancer remains controversial. Elevated serum levels of RBP4 have been associated with an increased risk of breast cancer, with higher levels found in patients with negative progesterone or estrogen receptors compared to those with positive receptors [10]. Metastatic breast cancer patients also show higher RBP4 levels than non-metastatic patients [33]. Conversely, other studies suggest that RBP4 may inhibit tumor growth, as low RBP4 expression in breast cancer tissues has been positively correlated with better prognosis [44]. The underlying mechanisms through which RBP4 affects tumor behavior are still unclear, necessitating further investigation into its precise role in cancer progression.

In contrast, RBP7 has been more extensively studied in the context of breast cancer. Its expression has been linked to breast cancer progression and has potential as a prognostic marker. Low RBP7 expression has been associated with poorer overall survival and disease-free survival in breast cancer patients. Moreover, reduced RBP7 expression correlates with tamoxifen resistance in ERα-positive breast cancer, particularly in luminal A subtype patients [15]. Based on our single-cell and TMA data, RBP7 is preferentially localized to vascular endothelial and luminal epithelial compartments in TNBC. RBP7 has been identified as an endothelium-specific PPARγ cofactor that maintains antioxidant, adiponectin-dependent signaling and vascular homeostasis, and as a PPARγ target in adipose tissue that regulates lipid storage and lipogenic programming [45]. In breast cancer, methylation-mediated silencing of RBP7 activates PPAR and PI3K/AKT signaling, whereas ectopic RBP7 expression suppresses AKT/SREBP1 activity and reduces fatty-acid accumulation in hormone receptor-positive cells [8]. Together with evidence that SREBP1-driven lipid metabolic reprogramming fuels TNBC growth, angiogenesis, and immune escape [46], these observations support a model in which endothelial and luminal RBP7 constitute a protective lipid-regulatory hub at the tumor–vascular interface. We hypothesized that RBP7-high endothelial cells, via PPARγ-dependent transcription, limit oxidative stress and fatty-acid flux to adjacent TNBC cells while secreting adiponectin and other anti-inflammatory mediators, whereas luminal RBP7 restrains AKT/SREBP1-dependent de novo lipogenesis. Loss or downregulation of RBP7 in either compartment would therefore be predicted to generate a leaky, pro-oxidant, lipid-enriched microenvironment that promotes TNBC proliferation, EMT, and therapeutic resistance.

ScRNA-seq technology was used to analyze the roles of RBP4 and RBP7 in TNBC. Our single-cell analysis results indicated that RBP4 was mainly expressed in endothelial cells. While RBP7 was primarily expressed in endothelial cells, it was also expressed at low levels in tumor epithelial cells, proliferating tumor cells, and luminal epithelial cells. We further analyzed single-cell data and identified five core genes associated with RBP7: TYMS, TK1, ASF1B, UBE2C, and PCLAF. Current research on the relationship between these core genes and RBP7 is limited. In a study on acute respiratory distress syndrome (ARDS), RBP7 was found to be closely associated with CD14 cells, and the high expression of TYMS in both CD14 and B cells suggests a synergistic relationship in regulating immune responses in ARDS [47]. The association of UBE2C with RBP7, alongside other upregulated genes such as TK1 and TYMS, underscores its potential role in hepatocellular carcinoma progression and highlights UBE2C as a critical target for therapeutic intervention in this context [48].

However, this study has some limitations. Firstly, the pan-cancer analysis in our study was mainly based on TCGA data, which are enriched for patients from specific ethnic and geographic backgrounds, and clinical characteristics such as stage, treatment exposure, and molecular subtype are not uniformly distributed or completely annotated, potentially causing biases. Secondly, our findings primarily stem from computational analysis of transcriptomic data; mechanistic in vitro functional assays, such as RBP7 knockdown or overexpression, were absent, necessitating future exploration of the functional mechanisms of RBPs through in vivo and in vitro experiments. Additionally, due to the retrospective nature of correlation analysis, no temporal or interventional inference can be made in our study. Future studies should include prospective cohorts and even randomized clinical trials where needed.

## 5. Conclusions

In summary, our findings highlight the multifaceted involvement of RBP family members in tumor progression and patient prognosis, underscoring the need for more in-depth studies. These results indicate that RBPs hold promise as prognostic biomarkers in certain malignancies. Notably, we demonstrate that RBP4 and RBP7 have particularly important clinical relevance in breast cancer.

## Figures and Tables

**Figure 1 cancers-17-03706-f001:**
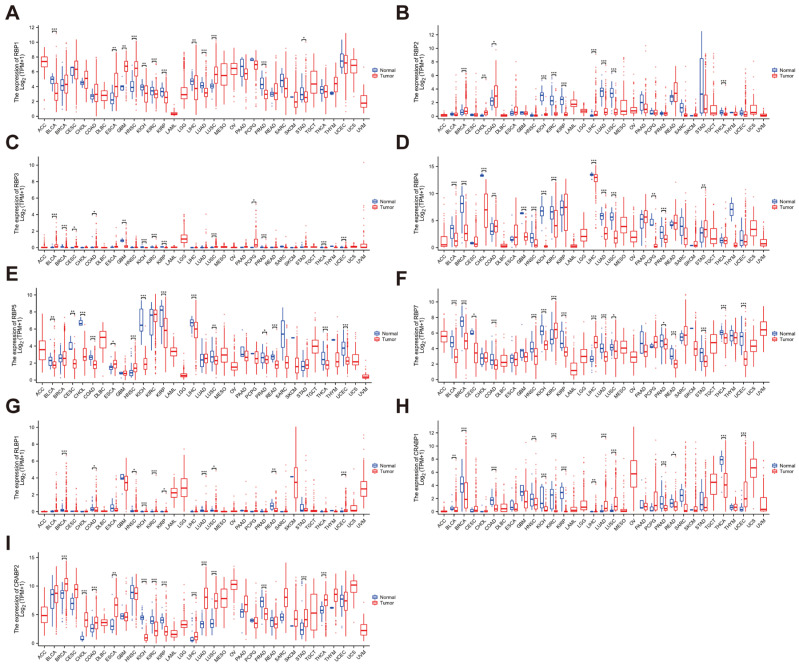
Differential expression of RBP family proteins across different cancer types. Boxplots of RBP family differential expression between cancer and adjacent normal tissues. Differential expression of (**A**) RBP1, (**B**) RBP2, (**C**) RBP3, (**D**) RBP4, (**E**) RBP5, (**F**) RBP7, (**G**) RLBP1, (**H**) CRABP1, and (**I**) CRABP2. The blue boxplots indicate the normal tissues. The red boxplots indicate the cancer tissues. * FDR < 0.05; ** FDR < 0.01; *** FDR < 0.001.

**Figure 2 cancers-17-03706-f002:**
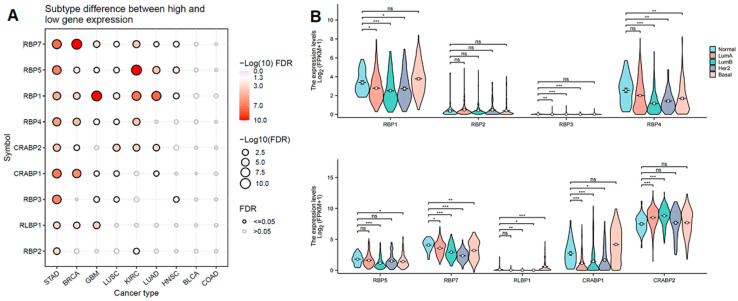
Expression of the RBP family in different subtypes. (**A**) Differential expression status of the RBP family for cancers with subtypes. Deeper color or larger circle size indicates greater significance. (**B**) Violin plot of differential expression status of the RBP family for breast cancers across different subtypes. * FDR < 0.05; ** FDR < 0.01; *** FDR < 0.001, ns = non-significant.

**Figure 3 cancers-17-03706-f003:**
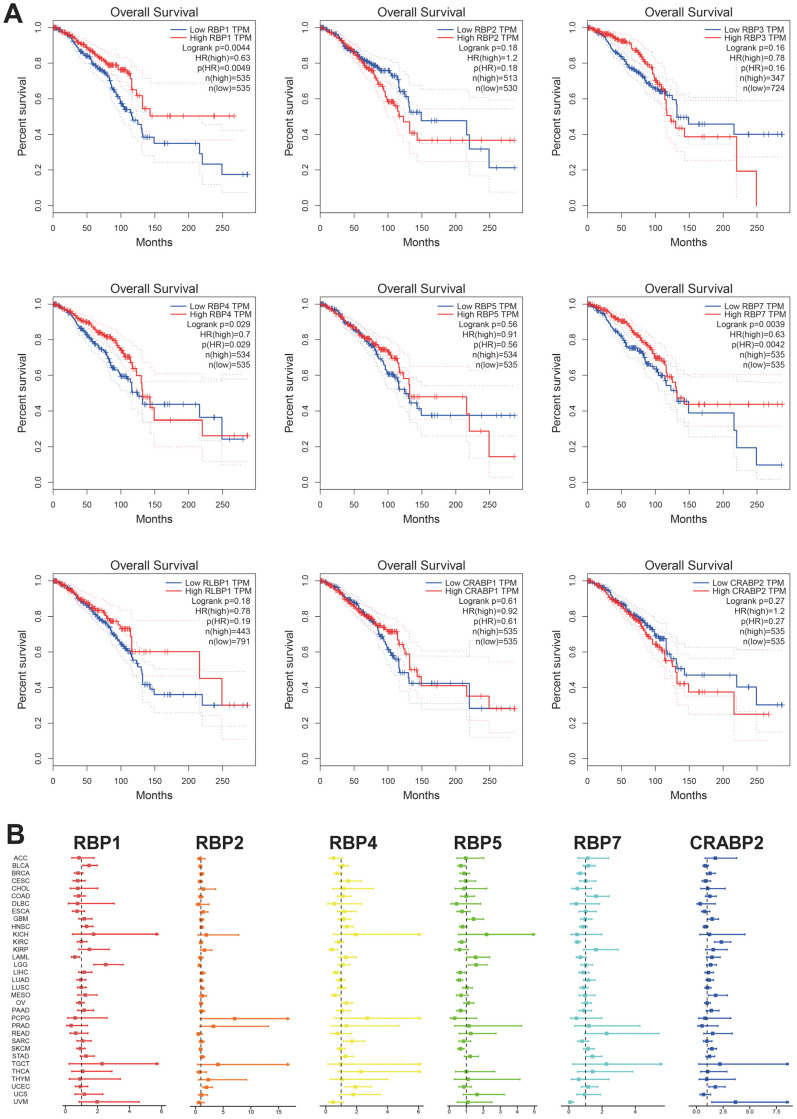
(**A**) Survival analysis of RBP family genes in BRCA. The red line in the graphs indicates high expression and the blue line indicates low expression. *p* value less than 0.05 is considered significant. (**B**) Correlation analysis of RBP family gene expression with survival by the Cox method in different types of cancers. Different colored lines indicate the risk value of different genes in tumors; hazard ratio < 1 represents low risk, and hazard ratio > 1 represents high risk.

**Figure 4 cancers-17-03706-f004:**
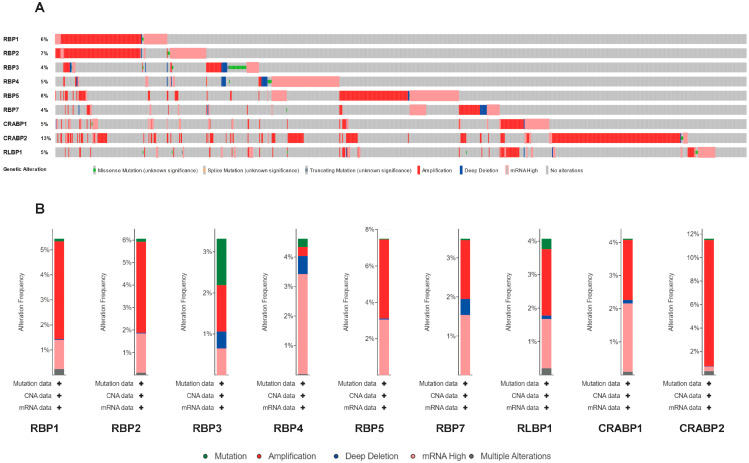
Genetic alterations and correlation analysis of RBP family members in pan-cancer. (**A**) Overview of alteration frequencies for RBP family genes in TCGA cohorts obtained via cBioPortal. (**B**) Genetic alteration frequency profiles for individual RBP family members across pan-cancer datasets. Different colors indicate different types of alterations, which are labeled below each figure.

**Figure 5 cancers-17-03706-f005:**
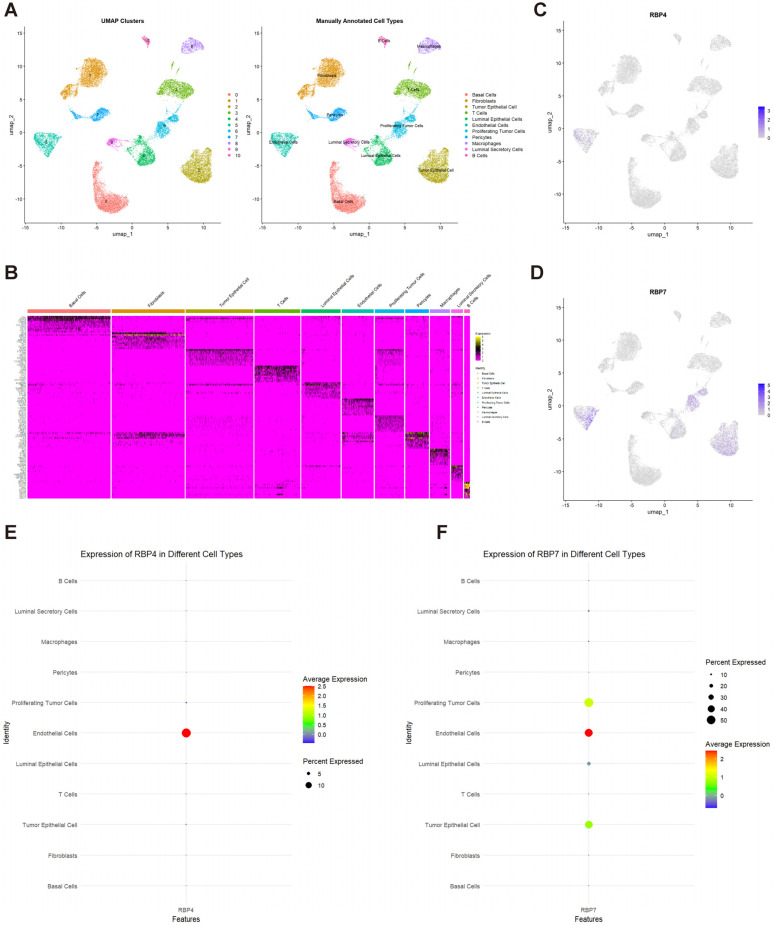
Cell cluster identification and RBP4/RBP7 expression in TNBC. (**A**) UMAP plot illustrating 11 manually annotated cell clusters. (**B**) Heatmap displaying the top 10 marker genes for each cell cluster. (**C**) UMAP plot for RBP4 expression in each cell type. (**D**) UMAP plot for RBP7 expression in each cell type. (**E**) Bubble plot for RBP4 expression for each cell type. (**F**) Bubble plot for RBP7 expression for each cell type. Deeper color in (**C**,**D**) indicates higher expression. Warmer color in (**E**,**F**) indicates higher expression, and a larger circle indicates a greater percentage of cells expressing these genes.

**Figure 6 cancers-17-03706-f006:**
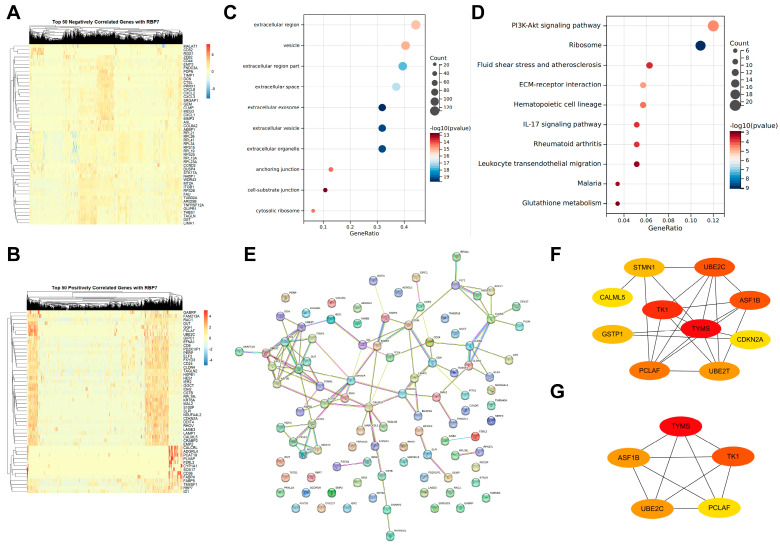
Gene correlation and enrichment analysis for RBP7 in tumor cells. (**A**) Heatmaps displaying the top 50 genes negatively correlated with RBP7 expression. (**B**) Heatmaps displaying the top 50 genes positively correlated with RBP7 expression. (**C**) GO functional enrichment analysis of differentially expressed genes. (**D**) KEGG pathway enrichment analysis. (**E**) Protein–protein interaction (PPI) network of the top 100 genes positively correlated with RBP7 expression. (**F**) Hub gene selection using cytoHubba, identifying the top 10 key genes. (**G**) Hub gene selection using cytoHubba, identifying the top 5 key genes. For (**F**,**G**), deeper color indicates the higher importance score as calculated in cytoHubba.

**Figure 7 cancers-17-03706-f007:**
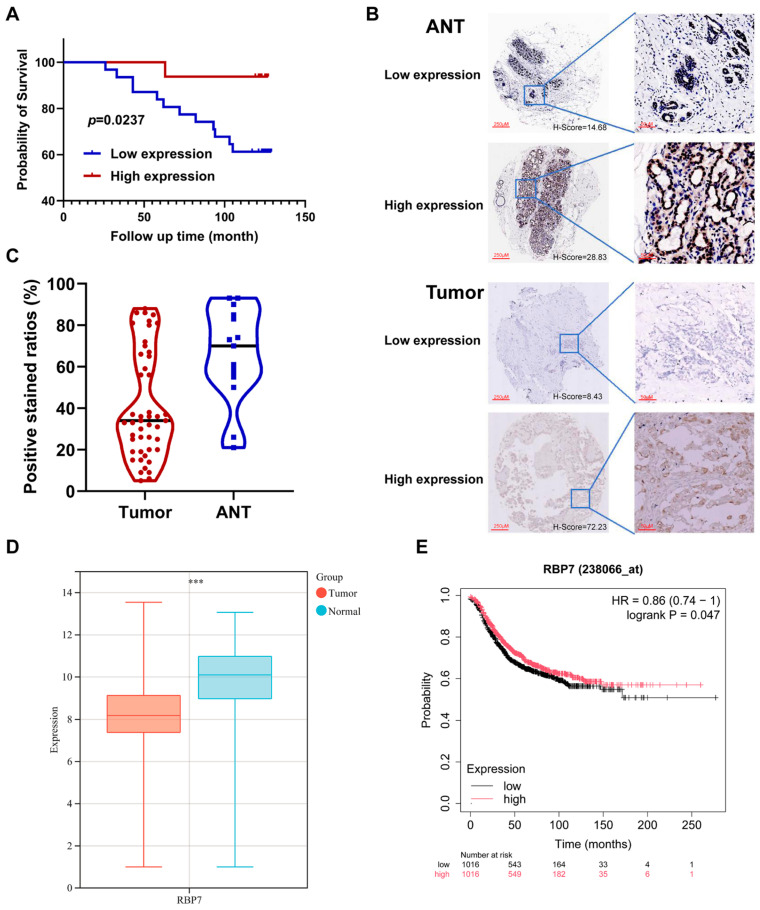
(**A**) Survival analysis of RBP7 in breast cancer patients. (**B**) Staining patterns of RBP7 in breast cancer tissue and adjacent non-cancerous tissue (ANT). Representative images show high and low expression of RBP7 in both tumor tissues and ANTs, along with corresponding magnified views of selected regions. (**C**) Statistical graph of RBP7 expression levels in breast cancer tissue and ANT. (**D**) Validation of RBP7 expression in breast cancer and normal tissues via GENT2. (**E**) Prognostic impact of RBP7 on breast cancer using the online tool “Kaplan–Meier Plotter-Breast cancer”. *** *p*< 0.001.

**Table 1 cancers-17-03706-t001:** Characteristics of 9 members of the RBP family in *Homo sapiens* using UniProt and GeneCards databases.

Gene Symbol	Ensembl	Protein Length (aa)	Subcellular Location	Qualified GO Term	Ligand	Oncogene	Tumor Suppressor Gene
RBP1	ENSG00000114115	135	Cytoplasm; lipid droplet	Is active in nucleus; located in nucleoplasm; cytoplasm; located in lipid droplet; located in cytosol	Retinol	Oral squamous cell carcinoma [26]	Ovarian cancer [27]; breast cancer [28]; endometrial cancer [29]
RBP2	ENSG00000114113	134	Cytoplasm	Is active in nucleus; cytoplasm; located in cytosol	Retinol		
RBP3	ENSG00000265203	1247	Extracellular matrix; secreted	Involved in retinoid metabolic process; involved in proteolysis; involved in lipid metabolic process; involved in visual perception	Retinoids		
RBP4	ENSG00000138207	201	Secreted	Retinoid binding; enables protein binding; enables retinal binding; enables retinol binding; enables retinol transmembrane transporter activity	Retinol	Glioblastoma [30]; ovarian cancer [31]; prostate cancer [32]; breast cancer [10,33]	
RBP5	ENSG00000139194	135	Cytoplasm	Enables retinoid binding; enables fatty acid binding; enables protein binding; lipid binding; enables retinal binding	Retinol		Hepatocellular carcinoma [34]
RBP7	ENSG00000162444	134	Cytoplasm	Retinoid binding; enables fatty acid binding; enables protein binding; lipid binding; enables retinal binding	Retinol		Breast cancer [15]
CRABP1	ENSG00000166426	137	Cytoplasm	Enables retinoic acid binding; enables retinoid binding; enables fatty acid binding; enables protein binding; lipid binding	Retinoic acid	Teratocarcinoma [35]; acute promyelocytic leukemia [36]; prostate cancer [37]; breast cancer [38]	Esophageal squamous-cell carcinoma [39]
CRABP2	ENSG00000143320	138	Cytoplasm; endoplasmic reticulum; nucleus	Enables retinoic acid binding; enables retinoid binding; enables fatty acid binding; enables protein binding; lipid binding	Retinoic acid; synthetic retinoid	Acute promyelocytic leukemia [40]; lung cancer [14]; pancreatic cancer [41]	Prostate cancer [42]; breast cancer [43]
RLBP1	ENSG00000140522	317	Cytoplasm	Enables 11-cis retinal binding; enables protein binding; enables retinol binding; enables phosphatidylinositol bisphosphate binding	11-cis-retinaldehyde/retinol/retinal		

## Data Availability

The datasets used in this study are available from the following sources: The Cancer Genome Atlas (TCGA) database and the Gene Expression Omnibus (GEO) database under accession number GSE161529. Additional data, such as tissue microarray data, are available from the corresponding author upon reasonable request.

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
