# Peer review of "Integrated Pan-Cancer Analysis and Experimental Verification of the Roles of Retinoid-Binding Proteins in Breast Cancer"

_cancers, 2025, doi:10.3390/cancers17223706_

Round 1

Reviewer 1 Report

Comments and Suggestions for Authors

This manuscript attempts to present an integrated pan-cancer analysis of retinoid-binding proteins (RBPs) with a focus on breast cancer, particularly triple-negative breast cancer (TNBC). While the topic may be of some biological interest, the study suffers from substantial conceptual, methodological, and interpretational weaknesses. The data analysis lacks rigor and originality, the biological validation is superficial, and the manuscript is poorly structured and redundant

Author Response

For research article

Response to Reviewer 1 Comments

1. Summary

Thank you very much for taking the time to review this manuscript. Please find the detailed responses below and the corresponding revisions in track changes in the re-submitted files.

2. Questions for General Evaluation

Reviewer’s Evaluation

Response and Revisions

Does the introduction provide sufficient background and include all relevant references?

Can be improved

Response: Thanks for your comments. We have improved the introduction section and made sure that it provided sufficient background and included all relevant references. The revised part was labeled with tracked-changes in the revised manuscript.

Are all the cited references relevant to the research?

Can be improved

Response: Thanks for your comments. We have made sure that all the cited references relevant to the research.

Is the research design appropriate?

Can be improved

Response: Thanks for your comments. We have improved the research design in our manuscript according to the comments made by all the reviewers and editors. The revised part was labeled with tracked-changes in the revised manuscript.

Are the methods adequately described?

Can be improved

Response: Thanks for your comments. We have described our methods adequately in our revised manuscript. The revised part was labeled with tracked-changes in the revised manuscript.

Are the results clearly presented?

Must be improved

Response: Thanks for your comments. We have improved the presentation of the results. The revised part was labeled with tracked-changes in the revised manuscript.

Are the conclusions supported by the results?

Can be improved

Response: Thanks for your comments. The conclusions of our study were supported by the results in the revised manuscript.

Are all figures and tables clear and well-presented?

Must be improved

Response: Thanks for your comment. We have improved the quality of all the figures and tables presented in our manuscript. For figures, we have improved the resolution of all figures to more than 600 ppi. For tables, we have revised all the tables according the comments of all reviewers and editors.

3. Point-by-point response to Comments and Suggestions for Authors

Comments: This manuscript attempts to present an integrated pan-cancer analysis of retinoid-binding proteins (RBPs) with a focus on breast cancer, particularly triple-negative breast cancer (TNBC). While the topic may be of some biological interest, the study suffers from substantial conceptual, methodological, and interpretational weaknesses. The data analysis lacks rigor and originality, the biological validation is superficial, and the manuscript is poorly structured and redundant

Response: Thanks for your comments. We have revised our manuscript substantially according to all the reviewers’ and editors’ comments. The revised part was labeled in tracked-change in the revised manuscript. For conceptual and methodological weaknesses, we have added the included validation cohorts based on data from ArrayExpress and GEO database (page 5, line 220 to line 226, and page 16, line 440 to line 448), conducted explicit quantification (percentage of RBP4+/RBP7+ cells per cluster) in single cell analysis section (page 5, line 198 to line 199, and page 14, line 398 to line 403). For interpretational weaknesses, we have made a clearer distinction between what is newly discovered in our study versus what has been previously established, and proposed a testable mechanistic hypothesis linking RBP7’s localization in endothelial and luminal cells with TNBC progression and microenvironmental regulation.

4. Response to Comments on the Quality of English Language

Point 1:NA

Response 1:    NA

5. Additional clarifications

NA

Reviewer 2 Report

Comments and Suggestions for Authors
  1. Overall Evaluation

This manuscript presents a comprehensive bioinformatics and experimental study examining the role of retinoid-binding proteins (RBPs), particularly RBP4 and RBP7, in cancer biology with a focus on triple-negative breast cancer (TNBC). Using large-scale TCGA datasets, cBioPortal mutation analyses, immune correlation profiling, and single-cell RNA-seq validation, the authors identify downregulation of RBP4 and RBP7 across multiple cancers and propose their potential as prognostic biomarkers. The study is well-motivated, methodologically detailed, and supported by experimental validation using tissue microarrays and immunohistochemistry.
However, certain aspects of the manuscript require refinement to enhance clarity, rigor, and interpretive depth before publication.

  1. Major Comments

2.1 Novelty and Significance

  • The integration of pan-cancer analysis with experimental validation adds translational relevance. However, the novelty is modest since prior studies have already reported associations between RBP7 and breast cancer outcomes (e.g., Lin et al., J Oncol 2022; Yan et al., Anti-Cancer Agents Med Chem 2023).
  • The manuscript would benefit from a clearer distinction between what is newly discovered here (e.g., the combined role of RBP4/RBP7, single-cell mapping, or prognostic implications across cancers) versus what has been previously established.

2.2 Study Design and Data Integration

  • The pan-cancer analyses and survival associations are comprehensive, yet the statistical rationale (e.g., adjustment for confounding variables, multivariate Cox regression) is insufficiently described. Clarify if age, stage, or molecular subtype were included as covariates in survival models.
  • Include a validation cohort (e.g., GEO dataset) to confirm key findings on RBP7 expression and prognostic impact, beyond TCGA.
  • The single-cell analysis section is methodologically solid but would benefit from explicit quantification (percentage of RBP4+/RBP7+ cells per cluster) and a statistical test of enrichment.

2.3 Biological Interpretation

  • The discussion adequately connects RBP7 with lipid metabolism (PPAR, PI3K/AKT, SREBP1 pathways), but mechanistic depth is limited. The authors could propose a testable mechanistic hypothesis linking RBP7’s localization in endothelial and luminal cells with TNBC progression or microenvironmental regulation.
  • The link between immune checkpoint expression and RBP expression is only briefly mentioned; expanding this with representative correlation plots (e.g., PD-L1, CTLA4, CD276) would strengthen the immuno-oncological significance.

2.4 Figures and Data Presentation

  • Figures are generally clear, but Figures 2 and 5 are repetitive and lack specificity. Ensure that each panel (A–F) is distinctly described.
  • Include statistical annotations (e.g., hazard ratio, p-value) directly on Kaplan–Meier plots.
  • The tissue microarray images should include magnified insets with consistent scale bars and staining intensity scoring (H-score or IRS).

2.5 Limitations

  • The authors briefly mention computational limitations. These should be expanded to include:
    • Potential biases inherent in TCGA sampling (ethnic and clinical heterogeneity).
    • Absence of mechanistic in vitro functional assays (e.g., RBP7 knockdown or overexpression in TNBC lines).
    • The retrospective nature of correlation analyses.
  1. Minor Comments
  1. Abstract: Consider shortening the background and emphasizing novel insights from pan-cancer and single-cell analyses.
  2. Language and Style: The manuscript is generally well-written, but some sentences are long and could be streamlined (e.g., lines 47–57, 187–195).
  3. Methodology: Specify the antibody catalog number and dilution used for RBP7 IHC.
  4. Statistical Analysis: Indicate whether p-values were corrected for multiple testing (e.g., FDR adjustment).
  5. Figure Quality: Improve resolution for Figure 3 and Figure 6 network plots.
  6. References: Reference [31] (Gabriel & Jatoi, 2012) seems contextually misplaced; consider verifying or replacing with a study directly related to RBP4 in breast cancer.
  1. Recommendation

Decision: Major Revision

The study is comprehensive and potentially impactful in defining RBP4 and RBP7 as prognostic biomarkers in breast cancer and other cancers. However, it requires:

  • Strengthening of novelty positioning,
  • Additional validation analysis, and
  • Enhanced discussion of biological mechanisms and limitations.

Addressing these concerns would significantly improve the manuscript’s clarity, reproducibility, and translational relevance.

Author Response

For research article

Response to Reviewer 2 Comments

1. Summary

Thank you very much for taking the time to review this manuscript. Please find the detailed responses below and the corresponding revisions in track changes in the re-submitted files.

2. Questions for General Evaluation

Reviewer’s Evaluation

Response and Revisions

Does the introduction provide sufficient background and include all relevant references?

Can be improved

Response: Thanks for your comments. We have improved the introduction to provide sufficient background and include all relevant references.

Are all the cited references relevant to the research?

Can be improved

Response: Thanks for your comments. We have made sure all the cited references relevant to the research.

Is the research design appropriate?

Can be improved

Response: Thanks for your comments. We have added additional analysis (such as the validation analysis) according the comments below.

Are the methods adequately described?

Can be improved

Response: Thanks for your comments. We have improved our methods according the comments below.

Are the results clearly presented?

Can be improved

Response: Thanks for your comments. We have made our results more clearly presented according to the comments below.

Are the conclusions supported by the results?

Can be improved

Response: Thanks for your comments. Our results supported the conclusions after revision according the detailed comments below.

Are all figures and tables clear and well-presented?

Can be improved

Response: Thanks for your comments. We have improved the presentation of figures and tables in our revised manuscript according the the comments below (such as improving the resolution of the figures)

3. Point-by-point response to Comments and Suggestions for Authors

Comments 1: Novelty and Significance: The integration of pan-cancer analysis with experimental validation adds translational relevance. However, the novelty is modest since prior studies have already reported associations between RBP7 and breast cancer outcomes (e.g., Lin et al., J Oncol 2022; Yan et al., Anti-Cancer Agents Med Chem 2023).

The manuscript would benefit from a clearer distinction between what is newly discovered here (e.g., the combined role of RBP4/RBP7, single-cell mapping, or prognostic implications across cancers) versus what has been previously established.

Response 1: Thank you for pointing this out. We agree with this comment. For Lin et al., they reported that lower expression of RBP7, the worse the prognosis in ER-positive (ER+) breast cancer patients. And for Yan et al. they reported that high expression of RBP7 was significantly related to good relative percent survival in the luminal A subtype, but in negative breast cancer, the result was opposite.

Different from these 2 previous studies, we primarily investigated the role of RBP7 in triple negative breast cancer (TNBC), and mapped both RBP4 and RBP7 in a single cell RNA dataset of TNBC. In addition, we also explored all the proteins in RBP family from a pan-cancer view. We displayed phylogenetic relationships, conserved protein motifs and gene structures of the RBP family members, investigated their expressions in all the cancer types in TCGA (Figure 1) and among different breast cancer subtypes (Figure 2), showed their prognostic implications across cancers, and explored genetic alterations and correlation analysis of RBP family members across cancers.

In conclusion, our study newly discovered the role of RBP4 and RBP7 in TNBC, and the role of proteins of RBP families across cancer types. We also highlighted the novelty and significance of our manuscript in page 18, line 477 to line 480.

Comments 2: Study Design and Data Integration: The pan-cancer analyses and survival associations are comprehensive, yet the statistical rationale (e.g., adjustment for confounding variables, multivariate Cox regression) is insufficiently described. Clarify if age, stage, or molecular subtype were included as covariates in survival models.

Include a validation cohort (e.g., GEO dataset) to confirm key findings on RBP7 expression and prognostic impact, beyond TCGA.

The single-cell analysis section is methodologically solid but would benefit from explicit quantification (percentage of RBP4+/RBP7+ cells per cluster) and a statistical test of enrichment.

  Response 2: Thanks for your comments. We used univariate COX regression considering that clinical covariates (age, stage, molecular subtype) are incomplete and heterogeneous across cancer types. Previous studies investigating the association between gene expressions and pan-cancer prognosis also applied univariate Cox regression1,2,3.

  We stated that we used univariate Cox regression while investigating the association between RBP expression and pan-cancer prognosis in the revised manuscript in page 4, line 162, and page 11, line 344.

  For validation cohort, we have used the GENT2 (http://gent2.appex.kr/gent2/), an online tool to investigate the gene expression profiles across tumor and normal tissues based on data from ArrayExpress database and GEO database, and also found a significant decrease of RBP7 expression level in breast cancer (Figure S5). We used the “Kaplan-Meier Plotter-Breast cancer” (https://kmplot.com/analysis/index.php?p=service&cancer=breast) online tool to investigate the prognostic impact of RBP7 also based on ArrayExpress database and GEO database, and also found that RBP7 sever as a protective factor for breast cancer. We updated the description of these validations in our revised manuscript in page 5, line 220 to line 226, and page 16, line 440 to line 448.

  In addition, we also calculated the percentage of RBP4+/RBP7+ cells per cluster and performed a statistical test of enrichment. The results were presented in our revised manuscript in page 5, line 198 to line 199, and page 14, line 398 to line 403 (Table S2 and Table S3, Figure S4 and Figure S5).

References:

1. Zhang, Y., Z. Li, H. Lu, Z. Jiang, Y. Song, Z. Ye, C. Gu, K. Chen, and A. Shang. "Pan-Cancer Analysis Uncovered the Prognostic and Therapeutic Value of Disulfidptosis." NPJ Precis Oncol 9, no. 1 (2025): 50.

2. Wang, Z. Q., Z. X. Wu, Z. P. Wang, J. X. Bao, H. D. Wu, D. Y. Xu, H. F. Li, Y. Y. Xu, R. X. Wu, and X. X. Dai. "Pan-Cancer Analysis of Nup155 and Validation of Its Role in Breast Cancer Cell Proliferation, Migration, and Apoptosis." BMC Cancer 24, no. 1 (2024): 353.

3. Zhang, C., J. Zheng, J. Liu, Y. Li, G. Xing, S. Zhang, H. Chen, J. Wang, Z. Shao, Y. Li, Z. Jiang, Y. Pan, X. Liu, P. Xu, and W. Wu. "Pan-Cancer Analyses Reveal the Molecular and Clinical Characteristics of Tet Family Members and Suggests That Tet3 Maybe a Potential Therapeutic Target." Front Pharmacol 15 (2024): 1418456.

Comments 3: Biological Interpretation: The discussion adequately connects RBP7 with lipid metabolism (PPAR, PI3K/AKT, SREBP1 pathways), but mechanistic depth is limited. The authors could propose a testable mechanistic hypothesis linking RBP7’s localization in endothelial and luminal cells with TNBC progression or microenvironmental regulation.

The link between immune checkpoint expression and RBP expression is only briefly mentioned; expanding this with representative correlation plots (e.g., PD-L1, CTLA4, CD276) would strengthen the immuno-oncological significance.

  Response 3: Thank you for pointing this out. In the revised manuscript, we have added a new paragraph in the Discussion (page 18 to page 19, line 506 to line 523) in which we propose a testable hypothesis based on our single-cell, bulk transcriptomic, and TMA data, together with recent literature. Specifically, we now discuss RBP7 as an endothelium-enriched PPARγ cofactor and lipid-regulatory molecule that may modulate oxidative stress, fatty-acid flux, and adiponectin-dependent signaling at the tumor–vascular interface, and as a negative regulator of PI3K/AKT–SREBP1–driven lipogenesis in luminal epithelial cells. We further hypothesize that loss of RBP7 in endothelial and luminal compartments could promote a lipid-enriched, pro-inflammatory microenvironment that favors TNBC growth, EMT, and therapy resistance.

For the link between immune checkpoint expression and RBP expression, we conducted co-expression analysis using the TCGA database to reveal the association between RBP4 and RBP7 with immune checkpoints in pan-cancer and showed the correlation plots for RBP4/RBP7 and all the immune checkpoints in Figure S2.

Comments 4: Figures and Data Presentation: Figures are generally clear, but Figures 2 and 5 are repetitive and lack specificity. Ensure that each panel (A–F) is distinctly described.

Include statistical annotations (e.g., hazard ratio, p-value) directly on Kaplan–Meier plots.

The tissue microarray images should include magnified insets with consistent scale bars and staining intensity scoring (H-score or IRS).

  Response 4: Thanks for pointing this out. For Figure 2, the legend has been changed into “Figure 2. The expression of RBP family proteins in different subtypes. (A) The differential expression status of RBP family proteins for cancers with subtypes. Deeper color or larger size of the circle indicating more significant. (B) Violin plot of differential expression status of RBP family proteins for breast cancers across different subtypes.”, and for Figure 5, the legend has been changed into “Figure 5. Cell Cluster Identification and RBP4/RBP7 Expression in TNBC. (A) UMAP plot illustrating 11 manually annotated cell clusters. (B) Heatmap displaying the top 10 marker genes for each cell cluster. (C-F) UMAP plot for RBP4 expression in each cell type. (D) UMAP plot for RBP7 expression in each cell type. (E) Bubble plot for RBP4 expression for each cell type. (F) Bubble plot for RBP7 expression for each cell type. Deeper color in (C) and (D) indicates higher expression. Warmer color in (E) and (F) indicates higher expression, and larger circle indicates more percentage of cells expressing these genes.”. We have ensured that each panel (A-F) is distinctly described, and ensured that all the figures were correctly cited in the manuscript.

For Kaplan-Meier plots (Figure 3A and Figure 7E), we have included the all the statistical annotations (e.g., hazard ratio, p-value) directly om each K-M plot.

For tissue microarray images, we included magnified insets with consistent scale bars and H-score (Figure 6).

Comments 5: Limitations: The authors briefly mention computational limitations. These should be expanded to include:

Potential biases inherent in TCGA sampling (ethnic and clinical heterogeneity).

Absence of mechanistic in vitro functional assays (e.g., RBP7 knockdown or overexpression in TNBC lines).

The retrospective nature of correlation analyses.

  Response 5: Thanks for your comments.We have expanded the limitation sections through involving “Potential biases inherent in TCGA sampling (ethnic and clinical heterogeneity), absence of mechanistic in vitro functional assays (e.g., RBP7 knockdown or overexpression in TNBC lines), and the retrospective nature of correlation analyses” (page 19, line 542 to line 554).

Comments 6: Abstract: Consider shortening the background and emphasizing novel insights from pan-cancer and single-cell analyses.

  Response 6: Thanks for your comments. In our revised manuscript, we have shortened the background and emphasized the novel insights from pan-can and single-cell analyses.

Comments 7: Language and Style: The manuscript is generally well-written, but some sentences are long and could be streamlined (e.g., lines 47–57, 187–195).

  Response 7: Thanks for pointing this out. We have streamlined line 47-57 (page 2, line 51 to line 65) and line 187-195 (page 6, line 240 to line 252). In addition, the manuscript was polished through Language and Figure Editing from Author Service, and we also uploaded the English editing certificate.

Comments 8: Methodology: Specify the antibody catalog number and dilution used for RBP7 IHC.

Statistical Analysis: Indicate whether p-values were corrected for multiple testing (e.g., FDR adjustment).

Response 8: Thanks for your comments. The antibody catalog number is 14541-1-AP (Proteintech) and the dilution is 1:400. For multiple testing (such DEG screening in pan-caner analysis, marker gene identification in scRNA-seq analysis), p-values were corrected using FDR adjustment.

Comments 9: Figure Quality: Improve resolution for Figure 3 and Figure 6 network plots.

Response 9: Thanks for your comments. We have improved the resolution of Figure 3 and Figure 6 to more than 1200 ppi and 900 ppi ,respectively.

Comments 10: References: Reference [31] (Gabriel & Jatoi, 2012) seems contextually misplaced; consider verifying or replacing with a study directly related to RBP4 in breast cancer.

Response 10: Thanks for your comments. We have replace the reference [31] (Gabriel & Jatoi, 2012) in the original manuscript by the reference “Jiao, C., L. Cui, A. Ma, N. Li, and H. Si. "Elevated Serum Levels of Retinol-Binding Protein 4 Are Associated with Breast Cancer Risk: A Case-Control Study." PLoS One 11, no. 12 (2016): e0167498.” and “Papiernik, D., A. Urbaniak, D. KÅ‚opotowska, A. Nasulewicz-Goldeman, M. Ekiert, M. Nowak, J. Jarosz, M. Cuprych, A. Strzykalska, M. Ugorski, R. Matkowski, and J. Wietrzyk. "Retinol-Binding Protein 4 Accelerates Metastatic Spread and Increases Impairment of Blood Flow in Mouse Mammary Gland Tumors." Cancers (Basel) 12, no. 3 (2020).” in the revised manuscript. And the new reference number is [10,33].

Comments 11: The study is comprehensive and potentially impactful in defining RBP4 and RBP7 as prognostic biomarkers in breast cancer and other cancers. However, it requires: 

Strengthening of novelty positioning,

Additional validation analysis, and 

Enhanced discussion of biological mechanisms and limitations.

Addressing these concerns would significantly improve the manuscript’s clarity, reproducibility, and translational relevance.

Response 11: Thanks for pointing all these drawbacks in our manuscript. These comments really helped us improve the quality of our manuscript. We have made strengthening of novelty positioning, additional validation analysis, and enhanced discussion of biological mechanisms and limitations according to the detailed comments listed above.

4. Response to Comments on the Quality of English Language

Point 1:NA

Response 1:    NA

5. Additional clarifications

NA

Reviewer 3 Report

Comments and Suggestions for Authors

Retinoic acid and its derivatives inhibit tumor cell growth in vitro. Retinoic acid preparations are used to treat acute promyelocytic leukemia and are being studied for their potential effects on other cancers. Given that breast cancer is one of the most common types of cancer worldwide, the search for new therapeutic solutions remains urgent. Retinoic acid preparations may offer such therapeutic solutions, and therefore, studying the role of retinoid-binding proteins in breast cancer meets the needs of modern fundamental oncology.

The study methods are consistent with the standards of the journal Cancers, are described in detail, and can be replicated should a similar study be needed for other cancer types or proteins.

The comments primarily relate to the presentation of the results. The article is well written and adequately reflects the results, but the figures and tables should be corrected.

Comments.

Line 191. Please check “uniport” or “UniProt”.

Line 192. "GO term"; both letters should be capitalized.

Table 1. It's a good idea to separate the rows in the table, as some rows merge and it's unclear which information refers to which RBP.

Figure 1 and Figure 2. Correct the figure captions. The captions should indicate what is shown in Figure 1A, Figure 1B, etc. The same applies to Figure 2. Also, in the caption for Figure 2, you wrote "boxplot," but in the figure there are "violinplots".

Section 3.4. Prognostic value of RBPs in pan-cancer. Figure 3. In Section 3.4, Figure 3B is mentioned first, then Figure 3A. I think it would be better to swap the parts of the figure: show the Kaplan-Meier curves in Figure 3A, and show the COX analysis results in Figure 3B.

Lines 363-364. Incorrect sentence: "In most cancers, including RBP1, RBP2, RBP3, RBP4, RBP5, RBP7, RLBP1, and CRABP1, RBPs exhibit downregulation." Correct the sentence to match what you intended to write. Perhaps it would be better to retranslate it into English.

Line 367. This line contains a typo: PRAD, not PRADZ.

Major comment.

Supplementary Materials are missing and therefore cannot be evaluated by the reviewer.

Comments on the Quality of English Language

The English should be made more academic, especially in the Simple Summary and Abstract.

I recommend using the English Editing Service to make the language more academic, as is customary in scientific publications.

Author Response

For research article

Response to Reviewer 3 Comments

1. Summary

Thank you very much for taking the time to review this manuscript. Please find the detailed responses below and the corresponding revisions in track changes in the re-submitted files.

2. Questions for General Evaluation

Reviewer’s Evaluation

Response and Revisions

Does the introduction provide sufficient background and include all relevant references?

Yes

Response: Thanks for your comments.

Are all the cited references relevant to the research?

Yes

Response: Thanks for your comments.

Is the research design appropriate?

Yes

Response: Thanks for your comments.

Are the methods adequately described?

Yes

Response: Thanks for your comments.

Are the results clearly presented?

Can be improved

Response: Thanks for point out this. We have made the results more clearly presented according to the comments below.

Are the conclusions supported by the results?

Can be improved

Response: Thanks for pointing out this. We have made that our conclusions were supported by the results in the revised manuscript according to the comments below.

Are all figures and tables clear and well-presented?

Must be improved

Response: Thanks for pointing out this. We have improved the presentation of all the figures and tables, such as improving the figure legend in Figure 1 and Figure2, improving the presentation of Table 1, and improving all the resolution of the figures in the revised manuscript.

3. Point-by-point response to Comments and Suggestions for Authors

Comments 1: Line 191. Please check “uniport” or “UniProt”.

Response 1: Thanks for your comments. We have changed the “uniport” into “UniProt” in page 6, line 245 and line 260

Comments 2: Line 192. "GO term"; both letters should be capitalized.

Response 2: Thanks for your comments. We have capitalized both letters (GO Terms) in the revised manuscript in page 6, line 247.

Comments 3: Table 1. It's a good idea to separate the rows in the table, as some rows merge and it's unclear which information refers to which RBP.

Response 3: Thanks for your comments. We separated the rows in table 1 in the revised manuscript.

Comments 4: Figure 1 and Figure 2. Correct the figure captions. The captions should indicate what is shown in Figure 1A, Figure 1B, etc. The same applies to Figure 2. Also, in the caption for Figure 2, you wrote "boxplot," but in the figure there are "violinplots".

Response 4: Thanks for your comments. We have updated the figure captions for Figure 1 and Figure 2 in our revised manuscript. For Figure 2, we corrected figure legend in addition to figure the figure caption.

Comments 5: Section 3.4. Prognostic value of RBPs in pan-cancer. Figure 3. In Section 3.4, Figure 3B is mentioned first, then Figure 3A. I think it would be better to swap the parts of the figure: show the Kaplan-Meier curves in Figure 3A, and show the COX analysis results in Figure 3B.

Response 5: Thanks for your comments. In the revised manuscript, we showed the Kaplan-Meier curves in Figure 3A, and show the COX analysis results in Figure 3B.

Comments 6: Lines 363-364. Incorrect sentence: "In most cancers, including RBP1, RBP2, RBP3, RBP4, RBP5, RBP7, RLBP1, and CRABP1, RBPs exhibit downregulation." Correct the sentence to match what you intended to write. Perhaps it would be better to retranslate it into English.

Response 6: Thanks for your comments. We corrected the expression of this sentence into “In the pan-cancer analysis, we found that RBP1, RBP2, RBP3, RBP4, RBP5, RBP7, RLBP1, and CRABP1 were down-regulated in the tumor group compared to the normal group across most of the cancer types in TCGA.” in the revised manuscript in page 18, line 463 to line 466.

Comments 7: Line 367. This line contains a typo: PRAD, not PRADZ.

Response 7: Thanks for pointing out this typo. We changed PRADZ into PRAD.

Comments 8: Supplementary Materials are missing and therefore cannot be evaluated by the reviewer.

Response 8: Thanks for your comments. We sincerely apologize for this oversight and thank the reviewer for pointing it out. In the revised submission, we have uploaded all Supplementary Materials (including the supplementary figures and tables) as separate files. We kindly invite the reviewer to evaluate these materials in support of the revised manuscript.

4. Response to Comments on the Quality of English Language

Point 1: The English should be made more academic, especially in the Simple Summary and Abstract.

I recommend using the English Editing Service to make the language more academic, as is customary in scientific publications.

Response 1: Thanks for your comments. We have polished our manuscript using the Language and Figure Editing from Author Service, and uploaded the English editing certificate.

5. Additional clarifications

NA

Round 2

Reviewer 1 Report

Comments and Suggestions for Authors

The response from the authors is satisfied. 

Reviewer 2 Report

Comments and Suggestions for Authors

The modified version is ready to be published